# Maternal Undernutrition Affects Fetal Thymus DNA Methylation, Gene Expression, and, Thereby, Metabolism and Immunopoiesis in Wagyu (Japanese Black) Cattle

**DOI:** 10.3390/ijms25179242

**Published:** 2024-08-26

**Authors:** Ouanh Phomvisith, Susumu Muroya, Konosuke Otomaru, Kazunaga Oshima, Ichiro Oshima, Daichi Nishino, Taketo Haginouchi, Takafumi Gotoh

**Affiliations:** 1Field Science Center for Northern Biosphere, Hokkaido University, N11W10, Kita, Sapporo 060-0811, Hokkaido, Japan; ouanh80@gmail.com (O.P.); haginouchi.taketo.b5@elms.hokudai.ac.jp (T.H.); 2Department of Animal Science, Joint Faculty of Veterinary Medicine, Kagoshima University, Korimoto 1-21-24, Kagoshima 890-8580, Kagoshima, Japan; muros@vet.kagoshima-u.ac.jp (S.M.); oshima@agri.kagoshima-u.ac.jp (I.O.); 3Joint Faculty of Veterinary Medicine, Kagoshima University, Korimoto 1-21-24, Kagoshima 890-8580, Kagoshima, Japan; otomaru@vet.kagoshima-u.ac.jp; 4Division of Year-Round Grazing Research, NARO Western Region Agricultural Research Center, 60 Yoshinaga, Ohda 694-0013, Shimane, Japan; tenpoint@affrc.go.jp; 5Graduate School of Bioresource and Bioenvironmental Sciences, Kyushu University, 744 Motooka Nishi-ku, Fukuoka 819-0395, Fukuoka, Japan; 0214daichi@gmail.com

**Keywords:** CE-FTMS, DNA methylation, fetal thymus, gene expression, immunity, maternal nutrition, metabolism, RNA sequencing, Wagyu cattle, WGBS

## Abstract

We aimed to determine the effects of maternal nutrient restriction (MNR) on the DNA methylation and gene expression patterns associated with metabolism and immunopoiesis in the thymuses of fetal Wagyu cattle. Pregnant cows were allocated to two groups: a low-nutrition (LN; 60% nutritional requirement; *n* = 5) and a high-nutrition (HN; 120% nutritional requirement, *n* = 6) group, until 8.5 months of gestation. Whole-genome bisulfite sequencing (WGBS) and RNA sequencing were used to analyze DNA methylation and gene expression, while capillary electrophoresis–Fourier transform mass spectrometry assessed the metabolome. WGBS identified 4566 hypomethylated and 4303 hypermethylated genes in the LN group, with the intergenic regions most frequently being methylated. Pathway analysis linked hypoDMGs to Ras signaling, while hyperDMGs were associated with Hippo signaling. RNA sequencing found 94 differentially expressed genes (66 upregulated, 28 downregulated) in the LN group. The upregulated genes were tied to metabolic pathways and oxidative phosphorylation; the downregulated genes were linked to natural killer cell cytotoxicity. Key overlapping genes (*GRIA1*, *CACNA1D*, *SCL25A4*) were involved in cAMP signaling. The metabolomic analysis indicated an altered amino acid metabolism in the MNR fetuses. These findings suggest that MNR affects DNA methylation, gene expression, and the amino acid metabolism, impacting immune system regulation during fetal thymus development in Wagyu cattle.

## 1. Introduction

Nutritional status significantly affects early developmental processes and thereby influences the growth trajectory and long-term health. The Developmental Origins of Health and Disease (DOHaD) hypothesis, initially proposed by Dr. David Barker in the late 20th century, was developed on the basis of epidemiologic evidence linking adverse prenatal and early postnatal conditions, such as maternal malnutrition and intrauterine growth restriction (IUGR), with a higher risk of metabolic diseases in later life, including insulin resistance, diabetes mellitus, and obesity [1]. During fetal development, an inadequate maternal intake of essential macronutrients (proteins, carbohydrates, and fats) and micronutrients detrimentally programs the fetal metabolism and development [2]. The DOHaD concept has significantly advanced our comprehension of how early-life experiences can influence health outcomes, initially focusing on chronic conditions, such as cardiovascular disease and metabolic syndrome in humans. Recent studies have broadened the concept to include the effects of maternal nutrition on fetal development and the subsequent generational productivity in experimental animal models [3]. The consequences of maternal undernutrition are influenced by factors such as the timing of challenges during gestation, the type and severity of nutritional restriction, the duration of exposure, and the species being studied. For instance, a 50% nutritional restriction during early to mid-gestation in sheep results in impaired fetal growth and reduced T4 secretion [4]. Similarly, a study of 60% nutrient restriction in Wagyu cattle demonstrated altered growth hormone secretion during the last trimester of pregnancy [5]. Furthermore, this level of maternal nutrient restriction throughout gestation led to a reduction in fetal growth and impaired fetal organ development [6]. Epigenetic analyses revealed that restricted maternal nutrition disrupts the expression of genes related to growth factors, such as IGFs and FGFs, in the muscle of the bovine fetus [7]. In addition, maternal nutrient restriction influenced DNA methylation and the expression of genes associated with the glucose and lipid metabolism in the fetal liver [8].

The thymus plays a crucial role in the development of the immune system through thymopoiesis, a process in which progenitor cells migrate to the thymus and undergo maturation into T lymphocytes. Previous research has demonstrated that IUGR disrupts the development and maturation of vital fetal organs, including the thymus, in various animal species. For instance, protein restriction throughout gestation in rats alters their thymic structure and T cell subpopulations [9]. In addition, energy restriction in fetal lambs during late pregnancy reduces the thymic size, alters its histologic structure, and affects the expression of immunity-related genes [10]. Epigenetic research has demonstrated that DNA methylation, histone modifications, and regulation by non-coding RNA mediate the long-term effects of early-life exposure to challenges to gene expression and function. These epigenetic modifications play a crucial role in the regulation of gene transcription and can be influenced by maternal nutrition [11].

The complex regulation of epigenetic modifications in the thymus plays a key role in the development of immune responses and the maintenance of immune function across the lifespan of an organism. Within this regulatory framework, DNA methylation plays a key role by significantly influencing patterns of gene expression involved in chromatin remodeling processes [12], which can serve as a hallmark of cancer [13]. Differentially methylated regions (DMRs), genomic regions which exhibit variations in methylation according to the cellular state, have been identified as promising biomarkers that could help elucidate the processes involved in organ development and function [14]. The maintenance of homeostasis within the thymic microenvironment is crucial for optimal T cell maturation and a robust immune repertoire. This microenvironment comprises a complex network of non-lymphoid cells, cytokines, chemokines, extracellular matrix elements, and matrix metalloproteinases, all of which contribute to thymocyte differentiation and maturation [15]. Previous studies have demonstrated the significance of mitochondrial reprogramming for the regulation of immune responses and the maintenance of cell viability under conditions of nutrient stress, have helped identify the mechanisms involved in immune regulation and viability during nutrient stress, and may provide insight into potential strategies to enhance immune function in nutrient-deprived environments [16].

There is limited understanding of immune system development in large ruminants during the fetal period, including producers of highly marbled beef cattle, such as Wagyu cattle. Cattle production plays significant roles in supplying a valuable protein source and as an economic driver. The survival of neonatal calves is crucial for the economic sustainability of cattle production, and a comprehensive understanding of the influence of epigenetic factors on the immune system is imperative to optimize calf management practices. In addition to diligent care to reduce the morbidity and mortality of calves, this knowledge is essential for the maintenance of sustainable livestock operations and global food security. In the present study, we aimed to investigate the epigenetic consequences of maternal nutrient restriction and, in particular, the association of the DNA methylation of genomic regions with gene expression and the metabolism, with a specific focus on the thymopoietic system in the Wagyu fetal thymus.

We followed established methodologies that have been outlined previously [6,7]. One group of pregnant cattle was fed 60% of their nutritional requirements, while the control group was fed 120%, in accordance with the Japanese Feeding Standards for Beef Cattle [17]. At 8.5 months of gestation, the fetuses were surgically retrieved, and samples of the thymus were obtained and promptly frozen and stored for subsequent analysis. Whole-genome bisulfite sequencing (WGBS) was used to characterize the DNA methylation patterns, RNA sequencing was used for gene expression profiling, and capillary electrophoresis-Fourier transform mass spectrometry (CE-FTMS) was used for the metabolomic assessment of the fetal thymic tissue.

## 2. Results

### 2.1. Effect of MNR on the Thymic Phenotype of the Calves

To investigate the effects of MNR on the physiologic adaptation of the fetal thymus, we initially measured the body and thymus masses of the calves and found substantially lower fetal body and thymus masses in the low-nutrient (LN) group than in the high-nutrient (HN) group (*p* < 0.01; Table 1). Furthermore, the ratios of the fetal thymus and total fetal masses were significantly lower in the LN group (*p* < 0.05; 0.7 and 0.5 in the HN and LN groups, respectively). Thus, fetal and thymic growth were impaired in the MNR group.

### 2.2. DNA Methylation Profile

#### 2.2.1. Landscape of DNA Methylation

For the WGBS analysis, we selected six bovine fetal thymuses: the three from the LN group that had the lowest body mass, and the three from the HN group that had the highest body mass, thereby aiming to exaggerate the potential differences between the groups. WGBS generated a total of 610 million and 623 million raw reads, respectively. After filtering the raw read data, 609 million and 622 million clean reads were obtained. Approximately 331 million and 351 million of these clean reads were successfully mapped to the reference genomes, with unique matches for the LN and HN groups, respectively. The mapping efficiency for the LN group ranged from 53.4% to 55%, and for the HN group, it ranged from 53.6% to 58.3% of the bovine genome, respectively (Appendix A). The overall methylation patterns of the two groups were visualized using a heatmap of the results of hierarchical clustering analysis (Figure 1).

#### 2.2.2. Pattern of DNA Methylation (DMR and DMG)

A total of 13,796 DMRs were identified for the fetal thymuses of the two groups (*q* < 0.01), encompassing 8869 DMGs. Of these, 4566 genes exhibited hypomethylation, and 4303 genes showed hypermethylation in the LN group compared to the HN group. The DMRs were predominantly situated in intergenic regions, comprising more than half of the total methylation percentage, followed by introns, intron-exon boundaries, and 1–5 kb regions around the genes, promoters, and exons, respectively (Table 2).

The chromosomal distribution of DMRs was characterized by higher abundance on chromosomes 3, 2, 1, and 4. Conversely, chromosomes 25, 24, 27, and X had the fewest DMRs, with respect to both hypo- and hypermethylation (Figure 2).

#### 2.2.3. Functional Enrichment of DMGs

To investigate the potential biological roles of DMGs, we used the Kyoto Encyclopedia of Genes and Genomes (KEGG) to identify relevant pathways. The top 20 pathways that differed (*q* < 0.0001) included the cAMP signaling pathway, the Rap1 signaling pathway, and the inflammatory mediator regulation of the TRP channels, all of which were represented by both hypomethylated and hypermethylated DMGs. Notably, Hippo signaling and the parathyroid hormone pathway were predominantly associated with the hypermethylated genes, whereas the hypomethylated genes were significantly associated with the cGMP-PKG signaling pathway, GABAergic synapses, and insulin secretion (Figure 3).

For the remaining pathways which significantly differed (*q* < 0.001), we excluded common pathway terms. Among the pathways associated with hypoDMGs were the Ras signaling pathway and insulin, GnRH, cortisol, oxytocin, and aldosterone secretion. In addition, hyperDMG was associated with Fc gamma R-mediated phagocytosis and the Wnt signaling pathway (Table 3 and Appendix A). Several DMGs are important regulators of metabolic processes, growth dynamics, and the responses to various stressors. These genes, which include *CREB1*, *ADCY1*, *SLC25A4*, *FGF10*, *IGF1R*, *CACNA1D*, *MAPK3*, *SOS1*, *PAK1*, and *STAT3*, are associated with diverse signaling pathways (Appendix A).

### 2.3. RNA Sequencing Profiles

#### 2.3.1. Differences in the Gene Expression of LN and HN Fetal Thymuses

The RNA sequencing analysis identified 94 DEGs (*q* < 0.05), of which 66 were upregulated and 28 were downregulated in the LN group compared to the HN group. The gene expression profile is shown as a heatmap, generated following the hierarchical clustering analysis of DEGs in Figure 4.

The DEGs exhibited high prevalences on chromosomes 7, 13, 17, and 29 for the upregulated genes and on chromosomes 19, 4, 1, 3, and 5 for the downregulated genes. Single upregulated genes were identified on chromosomes 7, 13, 11, 18, and 25, and chromosomes 4, 8, 12, and 26 exclusively harbored downregulated genes, as shown in Figure 5.

#### 2.3.2. Results of the Functional Enrichment Analysis of the DEGs

The KEGG pathway analysis revealed several significant terms (*q* < 0.10) related to the upregulated genes, encompassing pathways such as the glutathione metabolism, chemical carcinogenesis-reactive oxygen species, oxidative phosphorylation, and metabolic pathways in LN and HN fetal thymuses. In addition, the downregulated genes were related to pathways including the MAPK signaling pathway, the C-type lectin receptor signaling pathway, and natural killer cell-mediated cytotoxicity in LN and HN fetal thymuses. Among the upregulated genes, *MIF*, *GPX1*, *GPX4*, *NDUFS8*, *NDUFB10*, *PC*, *IDUA*, *ACAD8*, *CYC1*, and *GGT7* were associated with metabolic pathways. Specifically, *GPX1*, *GPX4*, and *GGT7* were found to be associated with the glutathione metabolism, and the *NDUFS8*, *NDUFB10*, and *CYC1* genes in LN fetal thymuses were linked to oxidative phosphorylation. The downregulated genes *NLK*, *MAP3K20*, and *PAK1* were associated with the MAPK signaling pathway in LN fetal thymuses. Furthermore, *PTPN11* and *PAK1* were associated with natural killer cell-mediated cytotoxicity and the C-type lectin receptor signaling pathway (Table 4).

### 2.4. Results of the Overlapping DMG/DEG Gene Analyses of DNA Methylation and Gene Expression

To further evaluate the genomic effects of MNR on fetal thymus development, we investigated the relationship between DNA methylation and genome-wide gene expression. The merging analysis of DMGs and DEGs identified 25 overlapping genes. The genomic distribution of the overlapping genes was consistent with the results of the WGBS analysis, with overlaps being frequently identified in intergenic regions, followed by introns, intron–exon boundaries, and regions within 1–5 kb of genes. However, no overlaps were found in promoter regions (Appendix A).

Figure 6 presents the top 16 significant terms (*q* < 0.10) derived from the KEGG pathway analysis. This list includes glutamatergic synapse, cAMP signaling, GABAergic synapse, the cGMP-PKG signaling pathway, and neutrophil extracellular trap formation. Within these pathways, a total of 11 genes were identified, notably including *CACNA1D*, *GRIA1*, and *SLC25A4*, which were upregulated and linked to multiple pathway terms. In addition, the downregulated genes, such as *ATF6*, *PAK1*, and *SLC38A1*, were each associated with two pathway terms.

### 2.5. Immunity-Related Genes Affected by MNR in the Fetal Thymus

To identify the immunity-related genes that are influenced by MNR, we focused on six genes that are known to be closely linked to the bovine immune response [18,19]: *CD4*, *CD8a*, *CD8b*, *CD205* or *Ly75*, *IFI16*, and *FOXN1*. The RNA-seq of these target genes in the LN and HN groups revealed a trend towards a higher expression of CD4 and CD8b in the LN group vs. the HN group, and the opposite trend for *IFI16* (*p* < 0.10). Subsequently, to validate these variations in gene expression, quantitative (q)PCR analysis was performed, which confirmed the differential expression (*p* < 0.05) of *CD4* (Appendix A).

#### Protein–Protein Interaction (PPI) Network Analysis of the Target Genes

PPI network analysis was conducted using STRING to elucidate the relationships among the six target genes. This identified eleven significant pathway terms (*q* < 0.05) across four categories: biologic processes (BP), two terms; cellular component (CC), one term; KEGG, six terms; and tissue development, two terms.

The genes *IFI16*, *CD8a*, *CD8b*, *CD4*, and *FOXN1* were linked with pathway terms associated with the regulation of the immune system (Figure 7). In addition, the BP T cell differentiation term was linked to *CD8a*, *CD4*, and *FOXN1*. The CC category was the cell surface term, and it was associated with *CD8a*, *CD8b*, *CD4*, and *LY75* (*CD205*). Notably, only *CD4* and *CD8b* were associated with the six pathway terms in KEGG. Within the tissue development category, two pathway terms were identified: the thymic epithelium term was associated with *CD4* and *FOXN1*, and the thymus term was associated with *CD8a*, *CD4*, and *FOXN1*. Moreover, CD4 and CD8 were associated with the primary immunodeficiency pathway in KEGG, which implies a potential link between MNR and defective immunity in the Wagyu fetal thymus (Appendix A).

### 2.6. Metabolomic Profile

To understand the effect of maternal undernutrition on the metabolism of the bovine fetal thymus, a comprehensive metabolomic analysis was conducted using CE-FTMS. A total of 500 peaks detected by mass spectrometry (MS) were identified as metabolites (Appendix A). Lidocaine, 2-deoxyadenosine, ethanolamine phosphate (all *p* < 0.01), histidine, ribulose 5-phosphate, 3-(3,4-dihydroxyphenyl)-2-methylalanine, alanine–alanine, serine, histamine, gluconolactone, 5-hydroxylysine, 1H-imidazole-4-propionic acid, N-acetylglucosamine 1-phosphate, and phenylalanine (all *p* < 0.05), as well as nicotinamide and allo-threonine (*p* < 0.10), were present at higher concentrations in the LN than in the HN fetal thymus. Conversely, UDP-glucose/UDP-galactose (*p* < 0.01), N-acetylmethionine, terephthalic acid, and deoxycytidine monophosphate (dCMP) (*p* < 0.05) were present at lower concentrations (Table 5).

Figure 8 shows the heatmap generated by hierarchical cluster analysis (HCA) of the top 50 metabolites. Of these, lidocaine, histidine, histamine, serine, phenylalanine, nicotinamide, ethanolamine phosphate (EAP), ethanolamine, aspartame, and CMP-N-acetylneuraminate were found to be more abundant in the LN group. Conversely, UDP-glucose/UDP-galactose (UDP-glc/UDP-gal), UDP-N-acetylglucosamine (UDP-GlcNAc), N-acetylmethionine, and deoxycytidine monophosphate (dCMP) exhibited higher abundance in the HN group. The results of HCA imply that these metabolites contribute to the differences between the thymuses of the two nutrient treatment groups. MSEA was performed using the top 50 metabolites, and this identified significant differences in the metabolic pathways, including the following: amino sugar and nucleotide sugar metabolism (*p* < 0.001); histidine metabolism; glycine, serine, and threonine metabolism; phenylalanine, tyrosine, and tryptophan biosynthesis (*p* < 0.01); pentose and glucuronate interconversion; glycerophospholipid metabolism; phenylalanine metabolism; and ascorbate and aldarate metabolism (*p* < 0.05) (Table 6).

## 3. Discussion

### 3.1. MNR Affects the Phenotype of the Fetal Thymus

There is a growing debate in the livestock industry regarding stress and its negative effects on animal welfare, as well as regarding the psychologic and physiologic responses of animals to conventional agricultural practices and advanced production technologies. Stress has various causes in animals that are both internal and external and affect their immune function, feed intake, fertility, and overall well-being [20]. Adequate nutrition during gestation is crucial for maternal health and fetal development, optimal growth, the prevention of congenital anomalies, and the long-term health of the offspring. In contrast, MNR can cause significant congenital anomalies by reducing the supply of vital nutrients, which reduces cellular proliferation and differentiation, and, therefore, fetal organ development [21].

In humans, placental insufficiency, leading to fetal growth restrictions, is linked to a lower birth weight and smaller visceral organs, including the thymus gland [22]. Research in Wistar rats has shown that nutrient restriction during the latter two-thirds of pregnancy results in smaller fetuses and a low fetal thymic mass, compared to fetuses from adequately nourished mothers [23]. Likewise, pregnant guinea pigs that experience nutrient restriction during pregnancy exhibit changes in fetal body composition and thymus mass [24]. In addition, nutrient-restricted pregnant ewes have been shown to produce offspring of small sizes, organ dimensions, and thymic mass [25]. The present findings are consistent with these previous observations made in various animal models, suggesting that MNR alters fetal composition and organ development [6]. In particular, we have identified the thymus as an organ that is particularly susceptible to the detrimental effects of nutritional deficiency.

### 3.2. MNR Affects DNA Methylation and Alters Cellular Function, as Shown by KEGG Pathway Analysis

In the present study, we aimed to investigate the effects of substantial MNR during pregnancy on the DNA methylation patterns within the fetal thymus. We identified significant alterations to the phenotype of the thymuses of the fetuses in the LN group, implying MNR-induced disruptions to genomic features which are crucial for thymic molecular and cellular functions. The analysis of DNA methylation patterns revealed diverse profiles, with the intergenic regions exhibiting the highest methylation levels, followed by the intronic regions, and lower methylation levels in the promoter regions, showing both hypo- and hypermethylation. These findings are consistent with the results of previous investigations conducted in bovine embryos [26]. Gene promoter methylation is conventionally thought to cause transcriptional suppression and the inhibition of gene expression [27]. While hypomethylation predominated in shared genomic regions, a different pattern was observed for the LN group’s promoters, in which the hypermethylated genes outnumbered the hypomethylated genes by nearly twofold (Table 2). Previous studies have shown that DNA methylation within the gene body may alter the chromatin structure and thereby influence transcriptional elongation [28]. In addition, DMRs were found to be more abundant in the intergenic and intronic regions, with higher frequencies of methylation being present on chromosomes 1, 2, 3, and 4 and lower frequencies being present on chromosomes 24, 25, 27, and X (Figure 2). The factors that influence these DNA methylation responses to environmental stimuli include gene density, developmental programming, nutrient-responsive genes, and chromatin accessibility [29]. The present findings suggest that chromosomes with a substantial number of DMGs exhibit high sensitivity to MNR in the thymuses of fetal Wagyu cattle.

The KEGG pathway analysis of the DMGs revealed several important upstream pathways among the top 20 significant pathway terms, including cAMP signaling, calcium signaling, and the growth hormone. The downstream pathways included Rap1 signaling, phospholipase D, and the glutamatergic synapse (Figure 3). These pathways play crucial roles in growth, metabolism, gene expression, synaptic plasticity, and neuronal function [30]. These upstream pathways may have significant effects on the metabolism, particularly under conditions of MNR. After excluding the top 20 terms and filtering out the shared pathways, the analysis revealed a 5.0-fold greater proportion of pathways that were associated with hypoDMG compared to hyperDMG (Table 3). This finding implies that MNR influences DNA methylation in the fetal thymus throughout gestation and is consistent with the findings of prior studies that maternal undernutrition modifies the DNA methylation profile of the bovine fetal liver [8].

The hypoDMG-associated pathways were involved in the regulation of cell proliferation, survival, differentiation, and apoptosis, including thymocyte development, such as Ras signaling. In murine studies, the Ras signaling pathway has been shown to be crucial for thymocyte development and T cell selection. This role is mediated via Ras guanine exchange factors, which influence the T cell repertoire at the second T cell receptor checkpoint [31]. In Ras-deficient mice, CD4 function is compromised, resulting in a reduction in the production of IFN-γ, a cytokine which is crucial for Th1 cell differentiation [32]. The cGMP–PKG signaling pathway is an important regulator of vascular tone, platelet function, and neuronal signaling, and it is also recognized for its involvement in the regulation of the actin cytoskeleton, which can be affected by nutrient stress [33]. Cortisol, the primary stress hormone, significantly affects the immune system. Chronic stress increases cortisol concentrations, which reduces the production of inflammatory mediators and cytokines, thereby affecting T cell activity [34]. In addition, the GABAergic synapse has been associated with hypoDMG, meaning that, therefore, inhibitory neurotransmitters and synaptic plasticity may also be affected by hypoDMG. The results further imply that the activation of GABAergic synapses is associated with the stress response, and the disruption of these pathways results in cellular dysfunction and an abnormal immune response [35]. HyperDMG was identified in association with the Hippo and Wnt signaling pathways, which are essential for the regulation of organ size, tissue regeneration, and cell proliferation during embryonic development [36]. The present findings suggest that MNR influences DNA methylation, thereby activating these pathways, in a process which would have affected the phenotypic outcomes of the fetus and thymus in the LN group. Furthermore, the parathyroid hormone (PTH) is important for calcium and phosphate homeostasis. The secretion of PTH during pregnancy is important to prevent vitamin D and calcium deficiency and to promote optimal fetal development [37]. In addition, the Fc gamma R (FcγR)-mediated phagocytosis of pathogens is crucial for the innate immune response. The activation of FcγRs is required for effector responses, endosomal maturation, and antigen processing [38], and the results of the present study imply that MNR may inhibit these pathways.

Several DMGs play crucial roles in the regulation of metabolic processes, growth dynamics, and stress responses. These regions are frequently located at the intron-exon boundaries and in introns and intergenic regions. Genes such as *SOS1* demonstrated hypomethylation in both introns and intergenic regions, *STAT3* and *FGF10* exhibited hypomethylation in the intronic regions, whereas *IGF1R* displayed hypomethylation at the intron-exon boundaries. *ADCY1* was characterized by hypomethylation in the intergenic regions, and *CREB1* displayed hypomethylation in the intergenic regions but hypermethylation in the introns. Similarly, *MAPK3* showed hypomethylation at the intron–exon boundaries and hypermethylation in the promoter region. These findings indicate that MNR affects the gene body, rather than the promoter region, in the thymuses of fetal Wagyu cattle. Conventionally, promoter methylation is thought to have larger effects on gene expression than the methylation of other genomic regions [27]. However, its effects may not be consistently linear because of complex interactions between the promoter and other genomic regions. For instance, the promoter hypermethylation of *MAPK3* was found to be linked to the FcγR pathway, and it may, therefore, impair immune responses associated with phagocytosis. Conversely, the *MAPK3* hypomethylation at the intron–exon boundary may affect cellular coordination via upstream pathways, such as the glutamatergic synapse, Ras signaling, and GnRH pathways. These findings suggest that MNR modifies DNA methylation patterns. Despite these methylation changes, no significant differences in the gene expression levels were detected, suggesting that these genes might remain transcriptionally inactive under MNR. In contrast, *SLC38A1* is crucial for the glutamine supply in immune cell proliferation and fetal development [39] and is linked to the glutamatergic and GABAergic pathways (Figure 6). Its hypermethylation and downregulation may impair protein synthesis and cell growth, potentially compromising thymocyte development, immune function, and overall fetal growth under nutrient restriction. This may represent a significant mechanism whereby fetuses adapt to survive under conditions of nutrient stress. However, such adaptations may also have consequences for the immunity of the calves.

### 3.3. MNR Regulates the Expression of Genes Associated with the Stress Response and Cellular Defense

The nutritional status of mothers has substantial effects on gene expression and thereby affects the tissue metabolism in the offspring pre- and postpartum. In the present study, we focused on measuring the effect of MNR throughout gestation on gene expression and extrapolated the biologic effects of altered expression in the fetal thymus. RNA-seq revealed a 2.4-fold larger number of upregulated rather than downregulated genes (66 vs. 28), and the subsequent KEGG pathway analysis showed a 4.0-fold higher number of dysregulated pathways associated with the upregulated genes compared to the downregulated genes in the LN group (Table 4). This suggests that MNR tends to upregulate, rather than downregulate, gene expression in the thymus of fetal Wagyu cattle in response to DNA methylation.

The downregulated genes, such as *NLK*, *MAP3K20*, *PTPN11*, and *PAK1*, are involved in several crucial pathways (Table 4). MAPK transduces extracellular signals and orchestrates essential cellular processes, including proliferation, differentiation, survival, and apoptosis. Previous studies have demonstrated that MAPK activation is associated with the upregulation of proinflammatory molecules, such as TNF-α [40]. Natural killer cell-mediated cytotoxicity plays a key role in immune responses against infections through direct cytotoxic effects [41]. In addition, the C-type lectin receptor pathway recognizes pathogens, triggering immune cell activation and cytokine production, and its dysregulation is implicated in various conditions, including cancer [42]. These pathways appear to be attenuated in LN fetuses, given that the expression of a number of genes expressing the components of these pathways was downregulated.

The functional activities associated with the upregulated genes were identified as nutrient recruitment and stress responses (Table 4). Metabolic pathways form complex networks of biochemical reactions that enable the acquisition, conversion, and utilization of energy from nutrients to support cellular processes. These pathways include genes such as *MIF*, *GPX1*, *GPX4*, *NDUFS8*, *NDUFB10*, *PC*, *IDUA*, *ACAD8*, *CYC1*, and *GGT7*, which are crucial for the glucose, protein, and lipid metabolism and cellular homeostasis. Maternal nutrient deficiency can cause intrauterine stress, leading to excessive reactive oxygen species (ROS) production and fetal oxidative stress, and organs that rely on oxidative phosphorylation (OXPHOS) are particularly affected. Metabolic pathways in the mitochondria are particularly important, such as the Krebs cycle, which oxidizes acetyl-CoA to produce NADH and ATP. OXPHOS is the primary mitochondrial pathway for ATP production, and it links the electron transport chain (ETC) with ATP synthase [43]. Under nutrient stress, cells undergo metabolic adaptation, using fatty acids and amino acids as alternative substrates for the Krebs cycle and OXPHOS. Thus, the appropriate regulation of OXPHOS is vital for the maintenance of cellular energy homeostasis and redox balance, the stress response, and cellular defense [44]. ROS, such as superoxide, hydrogen peroxide, and hydroxyl radicals, can damage DNA, lipids, and proteins, causing pregnancy complications. Nutrient deprivation disrupts mitochondrial function, increases ROS concentrations, triggers stress responses, and impairs cellular defenses. Glutathione (GSH) synthesis, which is crucial to limit the effects of ROS and maintain cellular redox balance as part of immune tolerance and the stress response, relies on amino acid availability [45]. Furthermore, antioxidants such as superoxide dismutase, catalase, and GSH peroxidase are important for oxidative balance. These stress response mechanisms help cells adapt to nutrient deprivation by modulating gene expression, reprogramming the metabolism, and enhancing cellular defenses, and deficiencies in these antioxidants can lead to oxidative stress and DNA damage [46]. The present findings imply that there may be abnormal interactions among the metabolic pathways, OXPHOS, ROS generation, and the GSH metabolism in the LN group, indicative of cellular stress in response to nutrient deprivation. The preservation of normal relationships among these pathways is crucial for cellular adaptation and survival when nutrients are scarce.

### 3.4. Metabolic Pathways Associated with Immune Function

Metabolite set enrichment analysis (MSEA) identified defects in several pathways that are linked to cellular defense through the regulation of the immune system in the LN group (Table 6). UDP-glucose, UDP-galactose, UDP-N-acetylglucosamine, and UDP-N-acetylgalactosamine are vital for immune cell function, enabling the production of glycoproteins and glycolipids. N-acetylglucosamine 1-phosphate is an intermediate in the biosynthesis of UDP-N-acetylglucosamine, and CMP-N-acetylneuraminic acid is the activated form of sialic acid [47]. These metabolites are linked to the amino sugar and nucleotide sugar metabolism, which is important for immune system regulation. The histidine metabolism generates histamine, a key signaling molecule in immune and inflammatory responses. Furthermore, essential derivatives such as 1-methyl-4-imidazoleacetic acid and imidazole-4-acetic acid are crucial for the regulation of histamine levels and signaling. Histamine plays a central role in inflammation [48]. The metabolism of glycine, serine, and threonine provides precursors and cofactors that are necessary for cellular processes, including immune system regulation. Serine, which is derived from 3-phospho-D-glycerate in the glycolytic pathway, is vital for protein, nucleic acid, and lipid synthesis, via conversion to glycine. Glycine can also be produced in mitochondria from threonine by threonine dehydrogenase and glycine C-acetyltransferase [49]. It is vital for the one-carbon metabolism and aids with cellular processes, including immune responses, through the glycine cleavage system [50].

During fetal development, threonine plays a role in metabolic adaptations to optimize nutrient usage and support organ development under conditions of poor maternal nutrition. Such adaptations involve an increase in the threonine concentration to meet fetal demands, which may have beneficial effects in organs such as the thymus. In the immune system, threonine interacts with a nearby amino acid in the FcγRIIB protein, causing a conformational change which affects the innate immune response [51]. Phenylalanine, tyrosine, and tryptophan are aromatic amino acids that are essential for the biosynthesis of important neurotransmitters and related hormones [52]. Similarly, tryptophan is a precursor for the synthesis of the neurotransmitter serotonin, the neurohormone melatonin, and the enzyme cofactors NAD and NADP [53]. This metabolic pathway may affect the immune system through effects on the synthesis of neurotransmitters, hormones, and cofactors. The pentose phosphate pathway and glucuronate interconversion are metabolic pathways that are important for the generation of NADPH and cellular redox homeostasis [54]. These pathways involve the interconversion of pentose sugars, such as ribulose 5-phosphate, and the production of glucuronic acid, which might be important for detoxification and, therefore, immune tolerance.

The glycerophospholipid metabolism is closely linked to fatty acid transport and the activity of enzymes such as long-chain acyl-CoA synthetase [55]. Nutrient deficiency alters the proportions of fatty acids in cell membrane glycerophospholipids and stimulates the breakdown of intracellular triglycerides, thereby increasing the glycerol and free fatty acid concentrations, which can affect glycerophospholipid synthesis and signaling. Disruptions to this metabolic pathway may have significant consequences for immune homeostasis [56]. Finally, glucuronic acid is a vital metabolite in the ascorbate and aldarate metabolism pathway, where it is involved in both the synthesis and breakdown of ascorbic acid (vitamin C) [57]. Furthermore, glucuronic acid aids in detoxification through glucuronidation reactions, thereby affecting immune responses. Disruptions to these pathways impair the cellular defense by compromising immune cell function and membrane integrity [58].

### 3.5. The Reprogramming of the Mitochondrial Metabolism Is Associated with Immune Competence

Pregnancy necessitates greater levels of protein synthesis to maintain pulmonary and systemic circulation and the maternal tissues and for fetal development. Essential nutrients, including glucose, amino acids, proteins, fatty acids, and cholesterol, support fetal processes, such as protein synthesis, substrate conversion, and oxidation. Amino acids are critical for protein synthesis and as precursors for compounds such as nitric oxide, polyamines, and glutathione, which are vital for placental growth, angiogenesis, and conceptus development [59]. Maternal nutrient restriction in sheep reduces the total α-amino acid concentrations, particularly those of serine, arginine family amino acids, and branched-chain amino acids, in the maternal and fetal plasma [60]. In the bovine fetus, maternal undernutrition alters the amino acid metabolism, resulting in high concentrations of histidine, phenylalanine, and glutamine in fetal muscle and of alanine and serine in the fetal liver [8,61]. Consistent with previous findings, in the present study, we identified high concentrations of several amino acids (histidine, serine, alanine, and phenylalanine) in the thymuses of the fetuses of the LN group. The thymus, a lymphoid organ which is crucial for immune regulation, requires amino acids for functions that are distinct from those in the muscle and liver. Specifically, amino acids are required in the thymus to provide energy for T cell production, which is essential for immune homeostasis and systemic protection.

Maternal nutrient deficiency may cause the fetal thymus to prioritize T cell development, with high amino acid concentrations indicating the presence of metabolic adaptations for ATP production via OXPHOS. This nutrient scarcity induces metabolic reprogramming, involving alterations to mitochondrial dynamics, an increase in oxidative stress, and a reorganization of the Krebs cycle, which is crucial for nutrient oxidation and the maintenance of cellular bioenergetics [43]. Mitochondrial OXPHOS is crucial for cellular energy synthesis, and converts ADP to ATP via the ETC, which is closely linked to the oxidative stress induced by excessive ROS production. This relationship is essential for metabolic adaptation, because the mitochondria adjust their function and metabolism in response to nutrient deprivation [62]. The amino acid metabolism is crucial for mitochondrial biogenesis, the antioxidant response, and the Krebs cycle. OXPHOS and the Krebs cycle are interconnected through the oxidation of NADH and FADH2 [63]. In addition, the mitochondria play a key role in the innate immune response by releasing ROS and other danger signals, such as mitochondrial DNA (mtDNA) and cardiolipin [64].

In the present study, we identified several hypoDMGs and upregulated genes that play significant roles in the upregulation of mitochondrial activity, secondary to increases in the concentrations of Krebs cycle intermediates. *SLC25A4* encodes the mitochondrial ADP/ATP carrier protein (ANT1), which facilitates ATP transport from the mitochondrial matrix to the cytoplasm and the import of ADP into the mitochondria, which is crucial for the cellular energy metabolism [65]. *CYC1* encodes a subunit of mitochondrial complex III that facilitates electron transfer for ATP generation in OXPHOS and the maintenance of the mitochondrial metabolism [63]. *NDUFS8* is a core subunit of mitochondrial complex I, which is a key component of the ETC, transferring electrons from NADH to ubiquinone for OXPHOS and ATP synthesis. Its role in proton pumping across the inner mitochondrial membrane is vital for ATP synthesis. The Krebs cycle is closely associated with this process, by providing NADH and FADH2 [66]. Similarly, *MRPS2* encodes mitochondrial ribosomal protein S2, which is crucial for ATP production via OXPHOS and the synthesis of proteins which are essential for the ETC [67].

*ACAD8* encodes an enzyme that is vital for mitochondrial fatty acid beta-oxidation. This process produces acetyl-CoA for the TCA cycle, which generates NADH and FADH2 [68]. Similarly, *PC* is a mitochondrial enzyme that is involved in the energy metabolism, specifically by facilitating the carboxylation of pyruvate, which replenishes oxaloacetate in the Krebs cycle, facilitating the supply of oxaloacetate to the TCA cycle and, thereby, the generation of NADH and FADH2, which are essential for ATP production through OXPHOS [69]. *CREB1* is a central regulator of genes that encode proteins involved in mitochondrial biogenesis and function, and it is recruited to the cytochrome c promoter to enhance biogenesis. The study of constitutively active– and dominant–negative *CREB1* variants has demonstrated their crucial role in the control of nuclear-encoded mitochondrial gene expression [70]. Furthermore, *CACNA1D* encodes Cav1.3, an L-type calcium channel subunit which regulates the calcium influx, including into the mitochondria. An increase in the mitochondrial calcium concentration mediated via Cav1.3 activates Krebs cycle enzymes, thereby enhancing NADH and FADH2 production [71]. Finally, *GRIA1* encodes a subunit of the AMPA receptor, a type of ionotropic glutamate receptor which plays a crucial role in synaptic transmission and neuronal excitability [72]. *GRIA1* interacts with *PGC1α*, a vital regulator of mitochondrial biogenesis. It also regulates *PPARα* and *PPARγ*, thereby influencing the glucose and lipid metabolism via the Krebs cycle and ATP production in the neural system [73].

Genes linked to pathways upstream of mitochondrial function may significantly affect the amino acid concentrations in the fetal thymus, thereby increasing energy generation by the Krebs cycle in the face of nutrient scarcity (Figure 9). Nutrient deprivation hampers the process of mitochondrial fission, reduces mitochondrial biogenesis, and improves survival through the facilitation of substrate sharing [74]. The oxidation of amino acids to generate essential metabolites by the Krebs cycle requires specific coenzymes derived from each type of amino acid. Previous studies have shown that histidine can be metabolized to glutamate by histidine ammonia-lyase and urocanate hydratase [75]. Glutamate can then be transaminated to form α-ketoglutarate, which is an intermediate in the Krebs cycle [76]. Serine can be converted to glycine and then to pyruvate through a series of enzymatic reactions [77], and pyruvate can be further metabolized to acetyl-CoA, which enters the Krebs cycle [78]. In addition, alanine can be converted to pyruvate by alanine aminotransferase [79]. Furthermore, the hydroxylation of phenylalanine to yield tyrosine is catalyzed by phenylalanine hydroxylase, which requires the cofactor tetrahydrobiopterin and molecular oxygen [80]. Moreover, threonine is first converted to α-ketobutyrate by threonine dehydratase. α-ketobutyrate is then converted to propionyl-CoA, which is further metabolized to succinyl-CoA. Succinyl-CoA then enters the Krebs cycle, where it is converted into intermediates used for energy production [81].

The cAMP-response element-binding protein (CREB), a key transcription factor, regulates gene expression in response to signals such as hormones, neurotransmitters, and growth factors. CREB-mediated signaling occurs within the mitochondria, as demonstrated by the discovery of a complete cAMP signaling pathway in the mitochondrial matrix [82]. Another study showed that CREB links Krebs cycle activity with OXPHOS and thereby regulates critical cellular processes, including glycolysis and immune checkpoint expression, which are essential for immune cell function [83]. In addition, CREB connects Krebs cycle activity, cellular energy status, and immune cell activation, and, thus, it regulates metabolic pathways and immune responses. CREB activation also induces regulatory T cell expression, thereby affecting proinflammatory responses [84]. Krebs cycle intermediates such as succinate, citrate, fumarate, and itaconate regulate cellular immunity and immune cell function. These metabolites act as signaling molecules, thereby modulating macrophage and T cell activity. Succinate and citrate are proinflammatory, and itaconate, α-ketoglutarate, and fumarate have other adverse effects. Specifically, they can affect immune cell epigenetics, thereby influencing polarization and innate immune dynamics [85].

In the present study, we identified the hypomethylation of the *CREB1* gene. The interplay between Krebs cycle intermediates, CREB1 signaling, and the regulation of mitochondrial function by calcium appears to fine-tune T cell functionality, thereby influencing the effects of CD4 and CD8 T cells (Figure 9). CD4 and CD8, vital glycoproteins on T cells, directly influence immune responses and form an important link between the cellular metabolism and immune function. CD4 helper T cells initiate primary CD8 T cell activation and guide these effector T cells to antigen-presenting cell (APC) sites through the production of chemokines and also maintain memory cytotoxic T cells [86]. Both CD4 and CD8 undergo unique developmental programs when activated, generating effector and memory T cell populations. However, the nature of the interaction between CD4 and CD8 cells is controversial. A mouse study demonstrated that low-protein diet-induced IUGR results in high CD4 expression in the fetal thymus [9]. However, high expression of CD4 disrupted the development of thymic epithelial cells, including the CD4 lineages [87]. CD4 activation is associated with inflammatory responses, which detrimentally affect the coordination of CD8 cells [26]. In the present study, we identified an upregulation of CD4 in the LN group, implying that this helper T cell may be involved in the primary immunodeficiency which characterizes nutrient restriction (Appendix A).

It is well established that the methylation of gene promoters is associated with transcriptional repression and the inhibition of gene expression. Conversely, methylation within gene bodies, including intronic regions, can enhance gene expression [27]. In the present study, we did not identify promoter-related regulation of gene expression. Instead, we identified complex methylation within gene bodies, including the intronic, intergenic, intron–exon boundary, and 1–5 kb regions. For instance, *GRIA1* was hypermethylated in its intronic and intergenic regions, whereas *SLC25A4* was hypermethylated at the intron–exon boundary, and both of these genes were upregulated. Conversely, *PAK1* displayed hypomethylation in the intronic and intergenic regions, and *SLC49A3* was hypomethylated at the intron–exon boundary, and both of these genes were downregulated (Appendix A). The KEGG pathway analysis revealed that the Hippo signaling pathway is linked to hyper-DMG, indicating that MNR affects the methylation of gene bodies, which may be responsible for the phenotypic outcomes of MNR identified in the present study. In addition, MNR induced alterations in the DNA methylation, gene expression, and metabolite composition of the fetal thymus. To maintain immune function under these conditions, the expression of genes related to mitochondrial activity, such as *MRPS2* and *SLC25A4*, is critical for the provision of energy with a minimal external nutrient input. Proper functioning of the thymus is essential for correct immune system function, and prolonged CD4 expression under these conditions may result in immune disorders in the affected animals later in life.

## 4. Materials and Methods

### 4.1. Experimental Animals and Diets

We studied 11 multiparous Japanese Black cows with similar body masses at the Iriki farm of Kagoshima University and the Western Region Agricultural Research Center, National Agriculture and Food Research Organization (NARO). They were treated in accordance with the Guide for the Care and Use of Experimental Animals, and the experimental design was approved by the Animal Care and Use Committee of Kagoshima University (approval number A18007). The animal management procedures followed previously established protocols [6]. In brief, pregnant cows were randomly assigned to an LN group (*n* = 5), which was fed 60% of the nutrient requirement, or an HN group (*n* = 6), which was fed 120% of the nutrient requirement. The nutrient calculations, including the energy requirements, were based on the standard diet per unit body mass, as per the JFSBC guidelines [17]. The composition of the diet, calculated on a dry matter (DM) basis, was 68.0% DM, 56.1% NDF, 36.0% ADF, 11.1% ash, 8.0% CP, 0.6% Ca, and 0.3% P. This diet has previously been described [7] and consisted of pelleted feed, total mixed ration, and rice straw. The experimental animals were individually fed in the morning (9 AM) and afternoon (4 PM). The original design included 6 animals per group, but we could not collect a sample from the LN group, resulting in a 6 vs. 5 distribution. This discrepancy is unlikely to affect data interpretation because the number of animals, despite being unequal, remains sufficient.

### 4.2. Thymus Sample Collection

After confirming pregnancy and allocating the cows to the experimental groups, the pregnancies were allowed to proceed until 8.5 months, when the fetuses were euthanized by exsanguination, following lidocaine injection into a jugular vein. Subsequently, samples were collected for the anatomic and physiologic analysis of the carcass and internal organs. Cesarean sections were conducted at the Veterinary Teaching Hospital of Kagoshima University. Thymus samples were collected, promptly frozen in liquid nitrogen, and stored at −80 °C until analysis. The samples were weighed, and an optimal volume was taken from the central part of the thoracic lobe (Lobus thoracicus) for each analysis.

### 4.3. DNA Methylation Analyses

#### 4.3.1. Whole-Genome Bisulfite Sequencing and Library Preparation

Genomic DNA was extracted from the thymuses of three fetuses from each of the LN and HN groups. The DNA methylation analysis was conducted by Rhelixa Co. Ltd., Tokyo, Japan. Genomic DNA was extracted from 30 mg frozen bovine fetal thymus tissue per sample using a Monarch Genomic DNA Purification Kit (New England Biolabs, Ipswich, MA, USA). The purity, concentration of nucleic acids, and degree of degradation of the isolated DNA were assessed using the NanoDrop One system (Thermo Fisher Scientific, Waltham, MA, USA). The DNA was quantified using Genomic DNA ScreenTape and Genomic DNA ScreenTape Sample Buffer, according to the protocol outlined in the Agilent Genomic DNA ScreenTape System Quick Guide (Agilent Technologies, publication number G2964-90040 rev.C, 2014). Libraries were generated according to the manual for the DNA Swift Biosciences™ Accel-NGS^®^ Methyl-Seq DNA Library Kit (Swift Biosciences, Inc., MI, USA). To ensure robust protection against DNA denaturation during WGBS analysis, we applied the EZ DNA Methylation-Gold Kit (Zymo Research, Irvine, CA, USA), which combines DNA denaturation and bisulfite conversion in a single step, enhancing conversion efficiency and reducing the processing time for high-throughput sequencing.

#### 4.3.2. Whole-Genome Bisulfite Sequencing Data Processing

A quality assessment of the raw paired-end sequence reads was conducted using FastQC v0.11.7. The trimming of low-quality bases (Phred score <20) or the first 10 bases from the 5′ end, as well as the removal of adapter sequences, was performed using TrimGalore (Version 0.5.0) and the options “-q 20 --phred33 --clip_R1 10 --clip_R2 10 –paired” [88]. The reads were aligned to the bovine reference genome (ARS-UCD1.2/bos, available online at https://www.ncbi.nlm.nih.gov/datasets/genome/GCF_002263795.1/, accessed on 15 October 2023), and processed using methylpy v1.4.6 and the options “--remove-clonal True --trim-reads FALSE” [89].

#### 4.3.3. Analyses of DMRs and DMGs

The tileMethylCounts function in the methylKit (Version 1.10.0) package was employed to estimate the number of methylated cytosines in 1000-base non-overlapping windows across the entire genome. Normalization was conducted using the normalizeCoverage function in methylKit. Bases with <10 read coverage and bases with coverage exceeding the 99.9th percentile in each sample were filtered out using the filterByCoverage function of methylKit. Subsequently, all the samples were merged using the unite function in the methylKit with the destrand = FALSE option [90]. DMRs were identified with *q* < 0.01, and regions showing a difference in methylation of >25% between groups were considered to be significantly differentially methylated. In instances where there was an overlap between a DMR and a specific gene functional element, the associated gene was classified as a DMG.

### 4.4. RNA Sequencing Analysis

#### 4.4.1. Sample Preparation for RNA-Seq Analysis

RNA was extracted from 30 mg of tissue per sample from the LN (*n* = 5) and HN (*n* = 6) groups using a TruSeq Stranded mRNA LT Sample Prep Kit. The extracted RNA underwent DNase I treatment to degrade double-stranded and single-stranded DNA, and the ribosomal RNA (rRNA) was sequenced using a NovaSeq 6000 S4 Reagent Kit, according to the manufacturer’s instructions (Illumina, Inc., San Diego, CA, USA). The RNA-seq analysis was performed by Macrogen Co. Ltd. (Tokyo, Japan).

#### 4.4.2. Quantification of Gene Expression and Differential Expression Analysis

Quality control of the data was performed using FastQC v0.11.7 [91], and raw sequences were subjected to quality trimming by Trimmomatic v.0.38 [92] prior to analysis to ensure data integrity. After quality control and raw data filtering, the clean reads were aligned to the bovine reference genome (ARS-UCD1.3/bos, accessible online at https://ftp.ncbi.nlm.nih.gov/genomes/all/GCF/002/263/795/GCF_002263795.2_ARS-UCD1.3/, accessed on 18 August 2023), using the fast spliced aligner in HISAT2 v.2.1.0 [93]. Bowtie2 v.2.3.4.1 was used to count the read numbers mapped to each gene [94]. Following read mapping, Stringtie v.2.1.7 was employed for transcript assembly, and the expression profile was determined for each sample and transcript/gene on the basis of the read counts.

Differential gene expression analysis of the LN and HN groups was performed using the DESeq2 package in R v.4.3.2 [95]. The *p*-values were adjusted using the Benjamini–Hochberg method [96], with a false discovery rate (*q*-value) threshold of <0.05 and a log2-fold difference of >1 required to accept significant differential expression. A hierarchical clustering analysis was conducted on the matrix using Euclidean distance as the distance metric and the complete linkage method to depict the expression patterns of genes and samples.

### 4.5. Gene Overlap Analysis

To characterize the relationships between the DEGs identified using the RNA-seq analysis and the DMGs identified using the DNA methylation analysis, shared genes were filtered using the dplyr package in R v.4.3.2.

### 4.6. qPCR Analysis

qPCR was conducted to compare gene expression using a CFX96 thermal cycler (Bio-Rad, Hercules, CA, USA) and a Thunder Bird SYBR qPCR kit (Toyobo Co., Ltd., Osaka, Japan), with the mRNA encoding ribosomal protein S18 (*RPS18*) as the reference gene. The evaluation of candidate gene expression was performed using the 2^−ΔΔCT^ method [97] and the one-sided Student’s *t*-test, with statistical significance being accepted when *p* ≤ 0.05 and a trend when *p* ≤ 0.10. The primers used in this study are listed in Appendix A.

### 4.7. Gene Functional Enrichment Analysis

To characterize the biologic functions of DMGs, DEGs, and overlapping genes, we performed a KEGG pathway enrichment analysis using the ShinyGO Bioinformatics database v.0.80 (http://bioinformatics.sdstate.edu/go/ [98]; accessed on 14 February 2024). In addition, to elucidate the roles of specific proteins in the biologic functions of target gene products, we analyzed PPI networks using the online database STRING v.12.0 (https://string-db.org/ [99]; accessed on 14 February 2024).

### 4.8. Metabolome Analysis

#### 4.8.1. Sample Preparation for CE–FTMS Analysis

Frozen thymus samples weighing 31–43 mg were suspended in 750 µL of a 50% acetonitrile solution (*v*/*v*) and a 2 µM internal standard solution from Human Metabolome Technologies (HMT), Tsuruoka, Japan. The mixture was then homogenized at 3500 rpm for 60 s, a step which was repeated 13 times at 0 °C to inhibit enzymatic activity. Subsequently, an equal volume of the 50% acetonitrile solution was added, before centrifugation at 2300× *g* and 4 °C for 5 min. The resulting supernatant was filtered through a Millipore 5 kDa cutoff filter using an ultrafiltration tube (Ultrafree MC PLHCC, HMT Inc., Tsuruoka, Japan). To eliminate macromolecules, the samples underwent further centrifugation at 9100× *g* and 4 °C for 120 min. The solvent was then allowed to evaporate and reconstitute in Milli-Q water for a subsequent metabolomic analysis at HMT.

#### 4.8.2. CE–FTMS Analysis

The CE–FTMS analysis was performed using MasterHands ver.2.19.0.2, an automatic integration software (Keio University, Tokyo, Japan), to identify peaks based on the signal/noise ratio (S/N > 3), the mass-to-charge ratio (*m*/*z*), the peak area, and the migration time (MT). The peak areas were normalized according to the method outlined by Arif et al. [100]. Putative metabolites were identified from HMT’s standard library and Known–Unknown peak library using MT and *m*/*z* with tolerances of ± 0.5 min and ± 0.5 ppm, respectively. In instances in which multiple peaks were associated with the same metabolite, branch numbers were assigned to differentiate them.

For the quantification of target metabolites, the calibration curve for the peak areas was adjusted using internal standard concentrations determined through a one-point calibration at 10 µM, with an internal standard concentration of 20 µM. When a compound was undetectable, its abundance was recorded as 0.

#### 4.8.3. Metabolomic Data Analysis

The effect of the maternal nutrition level on the metabolite concentrations was analyzed using two-sided Student’s *t*-tests in SPSS Statistics for Windows, version 25.0 (IBM Corp., Armonk, NY, USA). Differences were considered to be statistically significant at *p* ≤ 0.05 or to indicate a trend at *p* ≤ 0.10. The KEGG pathways were identified using MSEA in the online application MetaboAnalyst v.6.0 (https://www.metaboanalyst.ca/; accessed on 18 January 2024).

### 4.9. Hierarchical Clustering Analysis (Graphic Visualization)

To depict the patterns of methylation, gene expression, and metabolite concentrations, we conducted HCA using an online data visualization and graphing platform (http://www.bioinformatics.com.cn/en; accessed on 18 January 2024) [101]. The HCA was executed using a matrix with Euclidean distance as the distance metric and complete linkage as the method, facilitating the graphical classification of gene expression, metabolite profiles, and sample relationships.

## 5. Conclusions

In the present study, we comprehensively compared the DNA methylation patterns and gene expression profiles associated with alterations to the fetal thymus metabolism under MNR throughout gestation in Japanese Black (Wagyu) cattle. The functional analysis of DNA methylation and gene expression through a KEGG pathway analysis revealed that the MNR-induced modifications were predominantly hypomethylation and upregulation, rather than hypermethylation and downregulation. Both hypomethylation and hypermethylation were implicated in crucial pathways involved in the metabolism and growth regulation, including cAMP, calcium, Rap1, phospholipase D, and growth hormone signaling. The hypomethylated genes were associated with Ras signaling, the synthesis and secretion of insulin, and cortisol, and the hypermethylated genes were linked to the Hippo and Wnt signaling pathways, which may underpin poor fetal thymic growth, along with pathways related to immune regulation, such as Fc gamma R-mediated phagocytosis and parathyroid hormone signaling. The upregulated genes were associated with metabolic pathways, OXPHOS, chemical carcinogenesis via ROS, and the glutathione metabolism, and the downregulated genes were linked to MAPK signaling, C-type lectin receptor signaling, and natural killer cell-mediated cytotoxicity. In addition, alterations in the amino acid metabolism were identified, indicated by increases in the concentrations of histidine, serine, phenylalanine, alanine, and threonine. These modifications may contribute to characteristics of the immune system of the offspring in later life. Given the significant economic value of Wagyu cattle, further research is needed to elucidate the effects of MNR on postnatal immune system regulation and growth performance in the offspring.

## Figures and Tables

**Figure 1 ijms-25-09242-f001:**
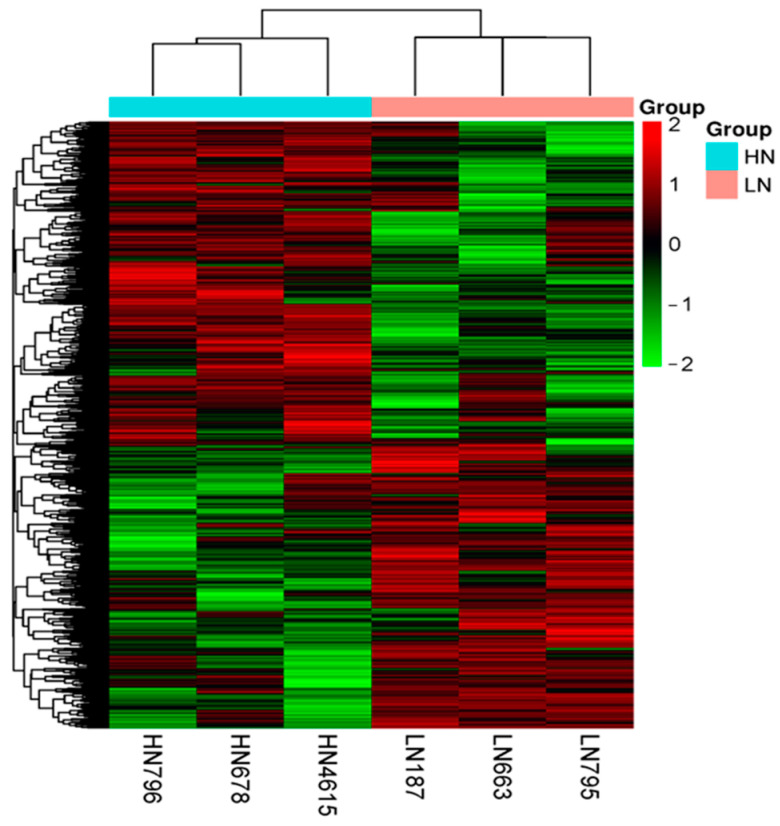
Heatmap showing the hierarchical clustering of differential methylation in the genomic regions for LN (salmon pink) and HN (green blue) fetal thymuses, comparing DNA methylation levels between LN and HN (*q* < 0.01). There were 4566 hypoDMGs and 4303 hyperDMGs in the LN group and the HN group. The row represents the gene, and the column denotes the sample. Red: hypermethylated genes; green: hypomethylated genes. The darkness of each color corresponds to the magnitude of the difference vs. the mean value.

**Figure 2 ijms-25-09242-f002:**
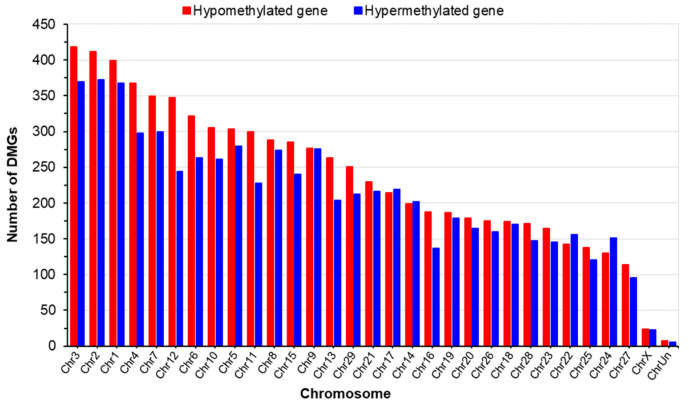
Chromosomal distribution of DMRs in the LN and HN thymuses. ChrUn, unplaced chromosome. Hypermethylation and hypomethylation DMGs in LN fetal thymuses are indicated in blue and red, respectively.

**Figure 3 ijms-25-09242-f003:**
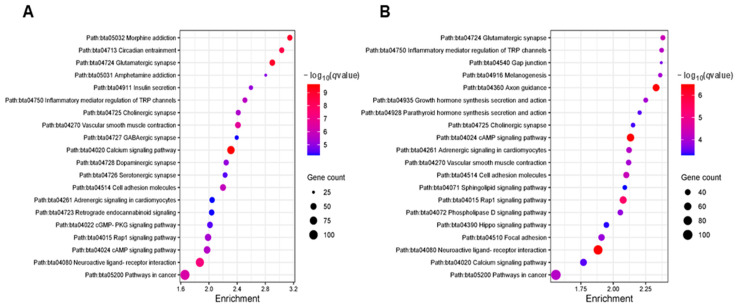
Scatter plot of KEGG pathway enrichment, depicting the top 20 significant pathways associated with DMRs. (**A**) Genes associated with all hypoDMRs. (**B**) Genes associated with all hyperDMRs. The *y*-axis represents the pathway terms, and the *x*-axis represents fold enrichment (the proportion of differentially methylated genes (DMGs) vs. all genes annotated with a specific pathway term). The size of the dots indicates the number of genes, and the color reflects the −log10(*q*-value).

**Figure 4 ijms-25-09242-f004:**
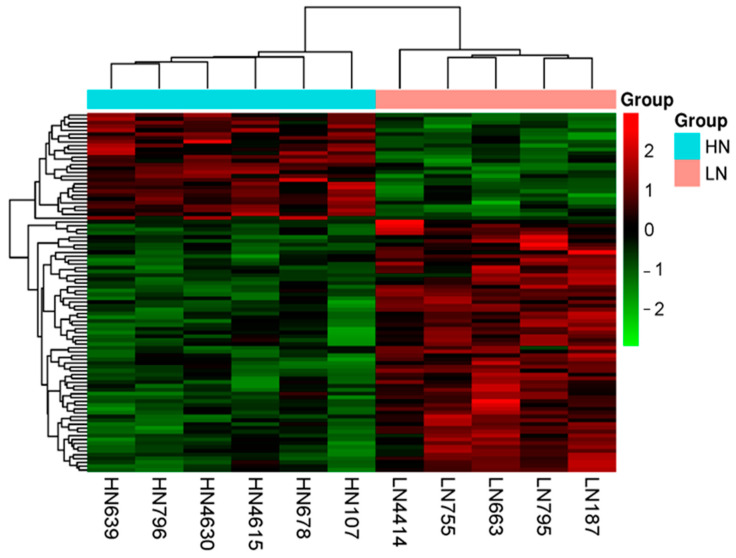
Results of the RNA sequencing analysis of the low-nutrition (LN) and high-nutrition (HN) groups. Heatmap of the hierarchical clustering of the 94 DEGs for LN (salmon pink) and HN (green blue) fetal thymuses (*q* < 0.05). The row represents the gene, and the column represents the sample. Red: upregulated genes; green: downregulated genes. The darkness of each color corresponds to the magnitude of the difference from the mean value.

**Figure 5 ijms-25-09242-f005:**
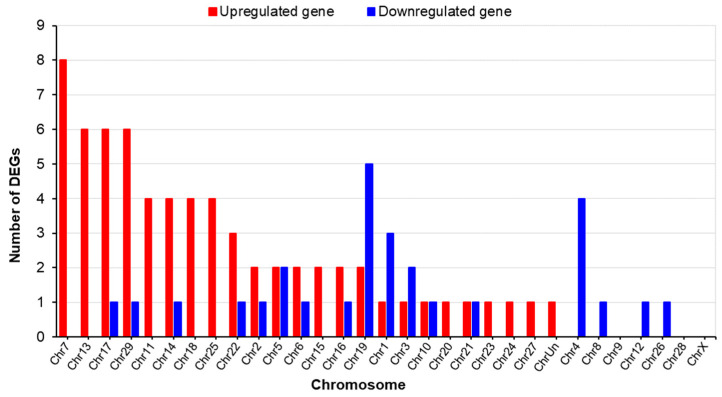
Chromosomal distribution of DEGs in the thymuses of LN and HN fetuses. ChrUn, unplaced chromosome.

**Figure 6 ijms-25-09242-f006:**
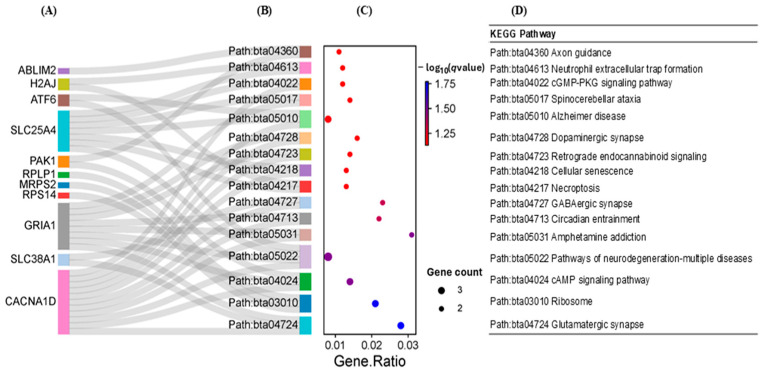
Sankey plot, illustrating the connections of overlapping genes identified using RNA-seq and DNA methylation analysis with the leading KEGG pathways. (**A**) denotes overlapping genes, obtained using DNA methylation and RNA-seq, that are linked to the KEGG pathways in (**B**). (**C**) The *y*-axis shows the names of genes and pathway terms, and the *x*-axis shows the ratio of the number of genes associated with terms to the number of genes in the pathway background. The dot size indicates the number of genes, and the color reflects the −log10(*q*-value). (**D**) ID and matched names for the KEGG pathway terms.

**Figure 7 ijms-25-09242-f007:**
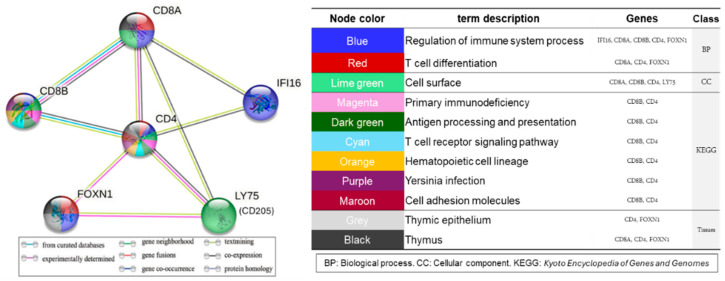
Results of the functional enrichment analysis on the protein–protein interaction (PPI) networks for six target genes. The node colors represent the enrichment of PPI pathways, with each color corresponding to a specific pathway. Significant enriched pathways (*q* < 0.05) are highlighted. BPs: biologic processes; CC: cellular component.

**Figure 8 ijms-25-09242-f008:**
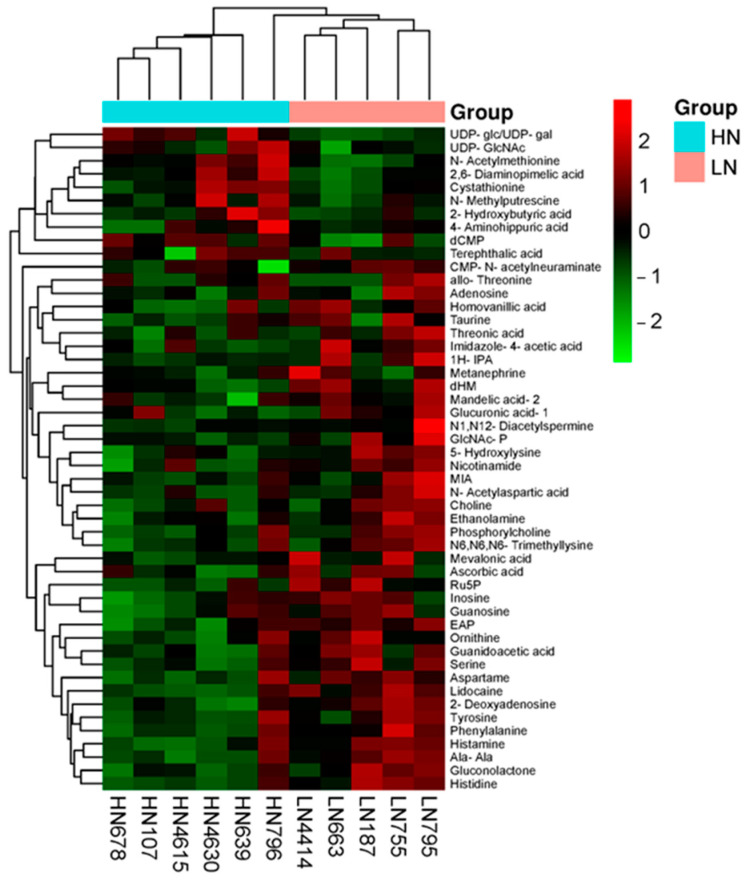
Heatmap of the hierarchical clustering of the top 50 metabolites present at differing concentrations in LN (salmon pink) and HN (green blue) fetal thymuses. The rows represent the metabolites, and the columns represent the samples. The metabolites present at relatively low concentrations are displayed in green, and those present at relatively high concentrations are displayed in red. The darkness of each color corresponds to the magnitude of the difference vs. the mean value. UDP-glc/UDP-gal, UDP-glucose/UDP-galactose; UDP-GlcNAc, UDP-N-acetylglucosamine; dCMP, deoxycytidine monophosphate; 1H-IPA, 1H-imidazole-4-propionic acid; dHM, 3-(3,4-dihydroxyphenyl)-2-methylalanine; GlcNAc-P, N-acetylglucosamine 1-phosphate; MIA, 1-methyl-4-imidazoleacetic acid; Ru5P, ribulose 5-phosphate; EAP, ethanolamine phosphate; and Ala-Ala, alanine–alanine.

**Figure 9 ijms-25-09242-f009:**
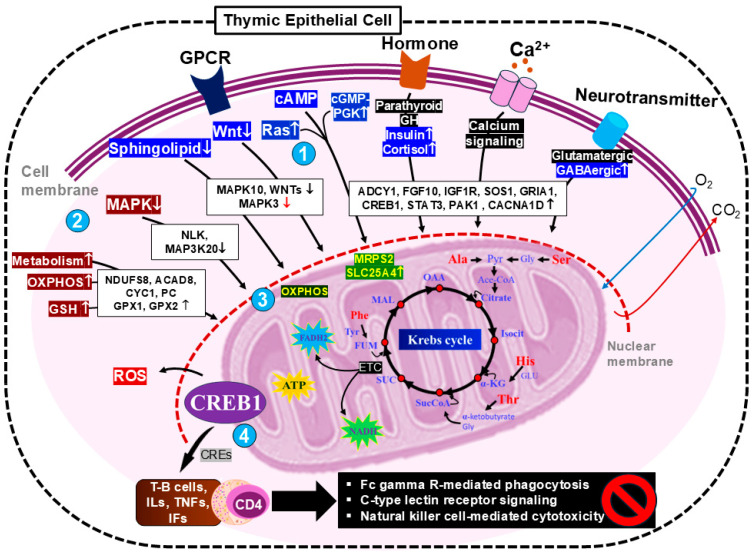
Theoretical framework comprising the pathways associated with the DMGs and DEGs associated with molecular function networks and metabolism in the thymuses of LN fetuses. (**1**) cAMP, Ras, cGMP–PGK, insulin, cortisol, and GABAergic synapse pathways, which are associated with hypoDMG, and Wnt and sphingolipid signaling pathways, which are associated with hyperDMG. (**2**) Pathways associated with upregulated genes, encompassing metabolic pathways, oxidative phosphorylation (OXPHOS), glutathione (GSH) metabolism, and reactive oxygen species (ROS). (**3**) Reprogramming of the mitochondrial metabolism via OXPHOS may be facilitated by MRPS2 and SLC25A4, with the oxidation of amino acids (His, histidine; Ser, serine; Ala, alanine; Phe, phenylalanine; and Thr, threonine) in the Krebs cycle generating ATP to support immune regulation. (**4**) cAMP-response element-binding protein 1 (CREB1) regulates energy production via mitochondrial OXPHOS and cAMP-response elements (CREs), thereby promoting immune cell development and, in particular, the generation of activated CD4 cells. Prolonged CD4 expression is associated with the suppression of immune pathways, including Fc gamma R-mediated phagocytosis, C-type lectin receptor signaling, and natural killer cell-mediated cytotoxicity. Up and down arrows indicate upregulation/hypomethylation and downregulation/hypermethylation, respectively, and the red arrows represent promoter hypermethylation. Pathways with a black background did not differ between the two groups.

**Table 1 ijms-25-09242-t001:** Effect of maternal nutrition on the phenotype of the fetal thymus.

Component.	HN (*n* = 6)		LN (*n* = 5)		Ratio	
	Mean	SEM ^1^	Mean	SEM ^1^	LN/HN	*p*-Value
Fetal age (d)	261.2	1.3	261.6	1.6	1.0	0.835
Fetal weight (g)	32,653.3	448.5	23,976.0	2540.9	0.7	0.005
Thymus (g)	212.0	21.2	110.5	22.2	0.5	0.009
% Body weight	0.7	0.001	0.4	0.001	0.7	0.046

^1^ SEM: standard error of the mean.

**Table 2 ijms-25-09242-t002:** Number of differentially methylated regions for each genomic region in LN fetal thymus ^1^.

Genomic Regions	Hypo		Hyper		Ratio (Hypo/Hyper)
	Count	%	Count	%	
Promoter	63	0.9	111	1.7	0.6
1–5 kb	278	3.8	253	3.9	1.1
Exon	55	0.8	76	1.2	0.7
Intergenic	3876	52.9	3317	51.2	1.2
Intron	2220	30.3	1891	29.2	1.2
Intron-exon boundary	764	10.4	765	11.8	1.0
None	65	0.9	62	1.0	1.0
Total	7321	100.0	6475	100.0	

^1^ Number of differentially methylated counts in the thymuses of LN and HN fetuses, separately indicated as hypermethylation or hypomethylation.

**Table 3 ijms-25-09242-t003:** Top significant pathways out of the top 20 KEGG pathways, extracted from hypo- and hypermethylated genes in the promoter, gene body, and/or UTRs ^1^.

Methylated Genes	KEGG Pathway	*q*-Value
Hypo		
	Path:bta04720 Long-term potentiation	0.00007
	Path:bta04360 Axon guidance	0.00010
	Path:bta04924 Renin secretion	0.00015
	Path:bta04072 Phospholipase D signaling pathway	0.00015
	Path:bta04927 Cortisol synthesis and secretion	0.00015
	Path:bta04921 Oxytocin signaling pathway	0.00018
	Path:bta04925 Aldosterone synthesis and secretion	0.00018
	Path:bta04971 Gastric acid secretion	0.00027
	Path:bta04540 Gap junction	0.00035
	Path:bta04912 GnRH signaling pathway	0.00035
	Path:bta04916 Melanogenesis	0.00038
	Path:bta04935 Growth hormone synthesis secretion and action	0.00039
	Path:bta04970 Salivary secretion	0.00039
	Path:bta05207 Chemical carcinogenesis-receptor activation	0.00039
	Path:bta05033 Nicotine addiction	0.00039
	Path:bta05414 Dilated cardiomyopathy	0.00039
	Path:bta04810 Regulation of actin cytoskeleton	0.00040
	Path:bta04730 Long-term depression	0.00041
	Path:bta04014 Ras signaling pathway	0.00050
Hyper		
	Path:bta04926 Relaxin signaling pathway	0.00081
	Path:bta04666 Fc gamma R-mediated phagocytosis	0.00070
	Path:bta05032 Morphine addiction	0.00070
	Path:bta05231 Choline metabolism in cancer	0.00070
	Path:bta04310 Wnt signaling pathway	0.00096

^1^ KEGG pathways obtained from DM genes in the LN fetal thymuses, separately indicated as related to hypermethylation or hypomethylation.

**Table 4 ijms-25-09242-t004:** Top significant differences (*q* < 0.10) in KEGG pathways, obtained from up- and downregulated genes in LN and HN fetal thymuses.

Direction	KEGG Pathway	*q*-Value	Genes
Upregulated			
	Path:bta03010 Ribosome	<0.0001	*RPS14*, *RPLP2*, *RPL37A*, *MRPS2*, *RPS15*, *RPLP1*, *RPS21*, *UBA52*, *RPL38*, *RPL28*, *RPL37*
	Path:bta05171 Coronavirus disease-COVID-19	<0.0001	*RPS14*, *RPLP2*, *RPL37A*, *RPS15*, *RPLP1*, *RPS21*, *UBA52*, *RPL38*, *RPL28*, *RPL37*
	Path:bta05022 Pathways of neurodegeneration—multiple diseases	0.002	*GPX1*, *SLC25A4*, *NDUFS8*, *NDUFB10*, *CACNA1D*, *CYC1*, *GRIA1*, *UBA52*
	Path:bta05016 Huntington disease	0.005	*GPX1*, *SLC25A4*, *NDUFS8*, *NDUFB10*, *CYC1*, *GRIA1*
	Path:bta00480 Glutathione metabolism	0.010	*GPX1*, *GPX4*, *GGT7*
	Path:bta04723 Retrograde endocannabinoid signaling	0.011	*NDUFS8*, *NDUFB10*, *CACNA1D*, *GRIA1*
	Path:bta04932 Non-alcoholic fatty liver disease	0.011	*NDUFS8*, *NDUFB10*, *CYC1*, *SREBF1*
	Path:bta05012 Parkinson disease	0.011	*SLC25A4*, *NDUFS8*, *NDUFB10*, *CYC1*, *UBA52*
	Path:bta05020 Prion disease	0.011	*SLC25A4*, *NDUFS8*, *NDUFB10*, *CACNA1D*, *CYC1*
	Path:bta05415 Diabetic cardiomyopathy	0.026	*SLC25A4*, *NDUFS8*, *NDUFB10*, *CYC1*
	Path:bta05014 Amyotrophic lateral sclerosis	0.029	*GPX1*, *NDUFS8*, *NDUFB10*, *CYC1*, *GRIA1*
	Path:bta05208 Chemical carcinogenesis-reactive oxygen species	0.032	*SLC25A4*, *NDUFS8*, *NDUFB10*, *CYC1*
	Path:bta05010 Alzheimer disease	0.032	*SLC25A4*, *NDUFS8*, *NDUFB10*, *CACNA1D*, *CYC1*
	Path:bta00190 Oxidative phosphorylation	0.047	*NDUFS8*, *NDUFB10*, *CYC1*
	Path:bta01100 Metabolic pathways	0.064	*MIF*, *GPX1*, *GPX4*, *NDUFS8*, *NDUFB10*, *PC*, *IDUA*, *ACAD8*, *CYC1*, *GGT7*
	Path:bta05031 Amphetamine addiction	0.078	*CACNA1D*, *GRIA1*
Downregulated			
	Path:bta04010 MAPK signaling pathway	0.077	*NLK*, *MAP3K20*, *PAK1*
	Path:bta04625 C-type lectin receptor signaling pathway	0.077	*PTPN11*, *PAK1*
	Path:bta05211 Renal cell carcinoma	0.077	*PTPN11*, *PAK1*
	Path:bta04650 Natural killer cell-mediated cytotoxicity	0.094	*PTPN11*, *PAK1*

**Table 5 ijms-25-09242-t005:** Top 20 metabolites present at differing concentrations in the thymuses of LN and HN fetuses.

Compound Name	HN (*n* = 6)		LN (*n* = 5)		LN/HN	
	Mean	SEM	Mean	SEM	*p*-Value	Ratio
UDP-glucose, UDP-galactose	0.01845	0.00148	0.01190	0.000449	0.004	0.6
2-Deoxyadenosine	0.00001	0.00000	0.00002	0.000001	0.010	1.3
Ethanolamine phosphate	0.22594	0.00462	0.24262	0.002295	0.014	1.1
His	0.03384	0.00152	0.04124	0.002097	0.017	1.2
N-acetylmethionine	0.00172	0.00010	0.00137	0.000074	0.020	0.8
Ribulose 5-phosphate	0.00150	0.00011	0.00203	0.000173	0.023	1.4
3-(3,4-Dihydroxyphenyl)-2-methylalanine	0.00013	0.00002	0.00024	0.000038	0.023	1.9
Ala-Ala	0.00152	0.00010	0.00188	0.000092	0.028	1.2
Ser	0.05118	0.00501	0.07142	0.006726	0.036	1.4
Histamine	0.00112	0.00018	0.00169	0.000143	0.039	1.5
Gluconolactone	0.00050	0.00003	0.00063	0.000046	0.044	1.2
5-Hydroxylysine	0.00099	0.00006	0.00124	0.000084	0.044	1.2
1H-Imidazole-4-propionic acid	0.00004	0.00001	0.00013	0.000041	0.047	2.9
N-acetylglucosamine 1-phosphate	0.00375	0.00010	0.00465	0.000424	0.050	1.2
Phe	0.12468	0.00845	0.15371	0.009901	0.051	1.2
Terephthalic acid	0.00016	0.00001	0.00011	0.000017	0.054	0.7
dCMP	0.00153	0.00007	0.00123	0.000124	0.054	0.8
Nicotinamide	0.05056	0.00397	0.06163	0.002909	0.058	1.2
Allo-threonine	0.000227	0.000033	0.000367	0.000041	0.064	1.6
Imidazole-4-acetic acid	0.000016	0.000003	0.000028	0.000005	0.064	1.7

**Table 6 ijms-25-09242-t006:** Metabolic pathways demonstrating the most significant differences in the thymuses of the LN and HN fetuses.

Pathway	Hits	Total	*p*-Value	Increased in LN Group	Decreased in LN Group
Amino sugar and nucleotide sugar metabolism	6	42	<0.001	N-Acetylglucosamine 1-phosphate and CMP-N-acetylneuraminic acid	UDP-N-acetylglucosamine; UDP-glucose; UDP-galactose; UDP-N-acetylgalactosamine
Histidine metabolism	4	16	0.001	His; Histamine; Imidazole-4-acetic acid; and 1-Methyl-4-imidazoleacetic acid	
Glycine, serine, and threonine metabolism	5	34	0.001	Serine; Choline; Allo-threonine; and 4-Guanidinoacetic acid	Cystathionine
Phenylalanine, tyrosine, and tryptophan biosynthesis	2	4	0.004	Phenylalanine and tyrosine	
Pentose and glucuronate interconversions	3	19	0.011	Ribulose 5-phosphate and Glucuronic acid	UDP-glucose
Glycerophospholipid metabolism	4	36	0.012	Choline phosphate; Choline; Ethanolamine phosphate; Ethanolamine	
Phenylalanine metabolism	2	10	0.025	Phenylalanine; and Tyrosine	
Ascorbate and aldarate metabolism	2	10	0.025	Glucuronic acid	UDP-glucose
Tyrosine metabolism	3	42	0.089	Tyrosine; Metanephrine; and Homovanillic acid	

## Data Availability

The data presented in this study are available on request from the corresponding author due to the data have not been published.

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
