# Peer review of "Maternal Undernutrition Affects Fetal Thymus DNA Methylation, Gene Expression, and, Thereby, Metabolism and Immunopoiesis in Wagyu (Japanese Black) Cattle"

_ijms, 2024, doi:10.3390/ijms25179242_

Round 1

Reviewer 1 Report

Comments and Suggestions for Authors

The manuscript titled “Maternal Undernutrition Affects Fetal Thymus DNA Methylation and Gene Expression, and thereby Metabolism and Immunopoiesis, in Wagyu (Japanese Black) Cattle” by Phomvisith, O.; et al. is a scientific work where the authors assessed how the dietary (high- vs low-nutrition, respectively) of Wagyu pregnant cows can impact on the DNA methylation and gene expression. Those specimens exposed to maternal nutrient restrictions exhibited DNA hypomethylation patterns that can alter the immune system regulation and lead to cancer diseases. This is a topic of growing interest. However, it exists some points that need to be addressed (please, see them below detailed point-by-point) to improve the scientific quality of the submitted manuscript paper before this article will be consider for its publication in the International Journal of Molecular Sciences.

1) The authors should consider to add the term “Wagyu cattle” in the keyword list.

2) “The consequences of maternal undernutrition (…) species being studied” (lines 51-54). What kind of nutrients are these previously reported works referring? Some insights should be furnished in this regard.

3) “Within this regulatory framework, DNA methylation plays a key role (…) cellular identity” (lines 76-78). Here, even if I agree with this statement provided by the authors it should be necessary to mention the importance of DNA methylation processes in the chromatinolysis process [1] that can serve as hallmark in cancer [2].

[1] Novo, N.; et al. Beyond a platform protein for the degradosome assembly: The Apoptosis-Inducing Factor as an efficient nuclease involved in chromatinolysis. PNAS Nexus 2022, 2, pgac312. https://doi.org/10.1093/pnasnexus/pgac312.

[2] O´Neill, H.; et al. Single-Cell DNA Methylation Analysis in Cancer. Cancers 2022, 14, 6171. https://doi.org/10.3390/cancers14246171.

4) Table 1 (line 127). Why the population size is different compared between high and low nutrient specimens, respectively? Could this negatively interfere during the subsequent data interpretation? Some information should be provided in this regard.

5) “The mapping efficiency for (…) 53.4% to 55% and from 53.6% to 58.3% (…) respectively” (lines 137-139). Please, the significant figures should be homogenized. This comment should be taken into account for the rest of the main manuscript body text.

6) “2.2. DNA Methylation Profile” (lines 129-194). Did the authors observe detrimental DNA degradation processes caused by the bisulfite treatment during the WGBS experiments? In case affirmative, what strategies were pursued in order to minimize this effect?

7) Conclusions (lines 828-847). This section perfectly remarks the most relevant outcomes found by the authors in this field work and the promising future perspectives. It should be desirable to add a brief statement to discuss about the potential future action lines to pursue the topic covered in this research.

Author Response

Reviewers’ comments:

Comments from Reviewer 1 to the Author:

Comment #1.

The authors should consider to add the term “Wagyu cattle” in the keyword list.

®Response: Thank you for your suggestions. We have added the term “Wagyu cattle” to the keyword list, and marked it in red and highlighted in yellow (line 35).

Comment #2.

“The consequences of maternal undernutrition (…) species being studied” (lines 51-54). What kind of nutrients are these previously reported works referring? Some insights should be furnished in this regard.

®Response: Thank you for your comment. We have revised the text to reference an experiment by Vonnahme et al. (2003) related to restricting 50% of global nutrient requirements in pregnant ewes during early to mid-gestation. The changes have been marked in red and highlighted in yellow (line 54-55: “For instance, a 50% nutritional restriction during early to mid-gestation in sheep results in impaired fetal growth and reduced T4 secretion [4].”).

Comment #3.

“Within this regulatory framework, DNA methylation plays a key role (…) cellular identity” (lines 76-78). Here, even if I agree with this statement provided by the authors it should be necessary to mention the importance of DNA methylation processes in the chromatinolysis process [1] that can serve as hallmark in cancer [2].

[1] Novo, N.; et al. Beyond a platform protein for the degradosome assembly: The ApoptosisInducing Factor as an efficient nuclease involved in chromatinolysis. PNAS Nexus 2022, 2, pgac312. https://doi.org/10.1093/pnasnexus/pgac312.

[2] O´Neill, H.; et al. Single-Cell DNA Methylation Analysis in Cancer. Cancers 2022, 14, 6171. https://doi.org/10.3390/cancers14246171.

®Response: Thank you for your suggestions. We have updated the text and added the citations. The changes have been marked in red and highlighted in yellow (line 79-81:” and the determination of cellular identity involved in chromatin remodeling processes [12] which can serve as a hallmark of cancer [13] “).

Comment #4.

Table 1 (line 127). Why the population size is different compared between high and low nutrient specimens, respectively? Could this negatively interfere during the subsequent data interpretation? Some information should be provided in this regard. This comment also related to comment from Reviewer B #6 mentioned that “Line 697. Why 5 animals and 6 animals?”

®Response: Thank you for your comment. We have addressed this in the Materials and Methods section. The original design included six animals per group, but we could not collect a sample from the LN group, resulting in a 6 vs. 5 distribution. This discrepancy is unlikely to affect data interpretation because the number of animals, despite the unequal distribution, remains sufficient (lines 712-715:” The original design included six animals per group, but we could not collect a sample from the LN group, resulting in a 6 vs. 5 distribution. This discrepancy is unlikely to affect data interpretation because the number of animals, despite being unequal, remains sufficient.”).

Comment #5.

“The mapping efficiency for (…) 53.4% to 55% and from 53.6% to 58.3% (…) respectively” (lines 137-139). Please, the significant figures should be homogenized. This comment should be taken into account for the rest of the main manuscript body text.

®Response: Thank you for your suggestions. We have updated the text and marked the changes in red and highlighted them in yellow (line 142-143: “The mapping efficiency for the LN group ranged from 53.4% to 55%, and for the HN group, it ranged from 53.6% to 58.3%” ).

Comment #6.

2.2. DNA Methylation Profile” (lines 129-194). Did the authors observe detrimental DNA degradation processes caused by the bisulfite treatment during the WGBS experiments? In case affirmative, what strategies were pursued in order to minimize this effect?

®Response: Thank you for your comment. We have addressed this concern in the Materials and Methods section, where it is marked in red and highlighted in yellow (line 738-741: “To ensure robust protection against DNA denaturation during WGBS analysis, we ap-plied the EZ DNA Methylation-Gold Kit (Zymo Research, Irvine, CA, USA), which combines DNA denaturation and bisulfite conversion in a single step, enhancing con-version efficiency and reducing processing time for high-throughput sequencing.”)

Comment #7.

Conclusions (lines 828-847). This section perfectly remarks the most relevant outcomes found by the authors in this field work and the promising future perspectives. It should be desirable to add a brief statement to discuss about the potential future action lines to pursue the topic covered in this research.

®Response: Thank you for your comment. We have included the suggestion to further investigate the effects of maternal nutrient restriction on long-term immune regulation and growth performance in the offspring. This has been marked in red and highlighted in yellow (line 864-866: “Given the significant economic value of Wagyu cattle, further research is needed to elu-cidate the effects of MNR on postnatal immune system regulation and growth performance in the offspring.”).

I would like to thank Reviewer A for your helpful comments and hope that the revised manuscript is acceptable for publication in the International Journal of Molecular Sciences.

Yours sincerely,
Takafumi Gotoh

Field Science Center for Northern Biosphere, Hokkaido University, N11W10, Kita, Sapporo 060-0811, Hokkaido, JAPAN

Tel & Fax: +81-11-706-3940

Reviewer 2 Report

Comments and Suggestions for Authors

It is a very complete work, the subject was addressed using cutting-edge techniques such as metabolomics, epigenetics, and transcriptomics.

However, some observations arose that are important to clarify to improve the manuscript

Line: 131: Were the analyses individual or a pool of three samples for each group?

Line: 151: And for the HN group?

Line 196: How many upregulated and downregulated samples for the HN group? Although it seems obvious, it is important to mention it

Line 259: Were the same RNA samples used for the RNAseq for validation?

Line 269: How were the genes IFI16, CD8a, CD8b, CD4, and FOXN1 expressed in the LN group?

Line 697. Why 5 animals and 6 animals?

Line 709: How much sample did they collect and what part of the thymus did they use?

Line 749: How much RNA did they use?

Line 775: Was the qPCR analysis to validate the differentially expressed genes?

Line 189: From the top 20 list, does any of them have a function with the immune system, which is related to the function of the thymus?

Line 828: In the discussion, it is imperative to mention which genes in the LN group were silenced by epigenetic mechanisms and were not significantly expressed, and how these genes affect the signaling pathway related to the immune system or the growth of the fetus.

Author Response

Comments from Reviewer 2 to the Author:

Comment #1.

Line: 131: Were the analyses individual or a pool of three samples for each group?

®Response: Thank you for your comment. We analyzed each sample individually.

Comment #2.

Line: 151: And for the HN group?

®Response: Thank you for your comment. We apologize for the lack of clarity regarding the comparison of DNA methylation between the two experimental groups. We compared the LN group to the HN group. Therefore, we have added the term “compared to the HN group” for better clarity (Line 155).

Comment #3.

Line 196: How many upregulated and downregulated samples for the HN group? Although it seems obvious, it is important to mention it.

®Response: Thank you for your comment. We apologize for the lack of clarity regarding the comparison of gene expression between the two experimental groups. The comparison was between the LN group and the HN group. Upregulation in the LN group corresponds to downregulation in the HN group, and vice versa. Therefore, we have added the term “compared to the HN group” for better clarity” (Line 202).

Comment #4.

Line 259: Were the same RNA samples used for the RNAseq for validation?

®Response: Thank you for your comment. Yes, we used the same RNA samples from the RNA-seq for validation.

Comment #5.

Line 269: How were the genes IFI16, CD8a, CD8b, CD4, and FOXN1 expressed in the LN group?

®Response: Thank you for your comment. We selected these genes based on the RNA-seq results to confirm their expression levels by qPCR analysis. This was done due to their roles in immune regulation, and the analysis compared expression levels in the LN group to those in the HN group individually (Figure S1).

Comment #6.

Line 697. Why 5 animals and 6 animals?

®Response: Thank you for your comment. We have addressed this in the Materials and Methods section. The original design included six animals per group, but we could not collect a sample from the LN group, resulting in a 6 vs. 5 distributions. This discrepancy is unlikely to affect data interpretation because the number of animals, despite the unequal distribution, remains sufficient (Line 712-714).

Comment #7.

Line 709: How much sample did they collect and what part of the thymus did they use?

®Response: Thank you for your comment. We collected the thymus from each animal and measured its weight. For each analysis, we used an optimal sample volume taken from the central part of the thoracic lobe (Lobus thoracicus) of the thymus. We have addressed this concern in the Materials and Methods section (Line 723-724; Line 729-730). We used 30 mg of frozen thymus tissue per sample for DNA extraction and applied at least 300 ng of DNA for WGBS analysis of each sample. The analysis procedures followed the specific protocol as described (Line 727-741:” Genomic DNA was extracted from thymuses of three fetuses from each of the LN and HN groups. The DNA methylation analysis was conducted by Rhelixa Co. Ltd., Tokyo, Japan. Genomic DNA was extracted from 30 mg frozen bovine fetal thymus tissue per sample using a Monarch Genomic DNA Purification Kit (New England Biolabs). The pu-rity, concentration of nucleic acids, and degree of degradation of the isolated DNA were assessed using the NanoDrop One system (Thermo Fisher Scientific, Waltham, MA, USA). The DNA was quantified using Genomic DNA ScreenTape and Genomic DNA ScreenTape Sample Buffer, according to the protocol outlined in the Agilent Genomic DNA ScreenTape System Quick Guide (Agilent Technologies, publication number G2964-90040 rev.C, 2014). Libraries were generated according to the manual for the DNA Swift Biosciences™ Accel-NGS® Methyl-Seq DNA Library Kit (Swift Biosciences, Inc., MI, USA). To ensure robust protection against DNA denaturation during WGBS analysis, we applied the EZ DNA Methylation-Gold Kit (Zymo Research, Irvine, CA, USA), which combines DNA denaturation and bisulfite conversion in a single step, enhancing conversion efficiency and reducing processing time for high-throughput sequencing.”).

Comment #8.

Line 749: How much RNA did they use?

®Response: Thank you for your comment. We extracted RNA from 30 mg of frozen thymus tissue for each sample and applied 1000-1300 ng of the extracted RNA per sample for analysis. The analysis procedures followed the specific protocol as described (Line 765-770:” RNA was extracted from 30 mg of tissue per sample from the LN (n = 5) and HN (n = 6) groups using a TruSeq Stranded mRNA LT Sample Prep Kit. The extracted RNA un-derwent DNase I treatment to degrade double-stranded and single-stranded DNA, and the ribosomal RNA (rRNA) was sequenced using a NovaSeq 6000 S4 Reagent Kit, according to the manufacturer’s instructions (Illumina, Inc., San Diego, CA, USA). The RNA-seq analysis was performed by Macrogen Co. Ltd. (Tokyo, Japan).”)

Comment #9.

Line 775: Was the qPCR analysis to validate the differentially expressed genes?

®Response: Thank you for your comment. After a comprehensive analysis of gene expression by RNA-seq, we employed qPCR analysis to further confirm the expression of the six target genes as mentioned in Comment #5 only.

Comment #10.

Line 189: From the top 20 list, does any of them have a function with the immune system, which is related to the function of the thymus?

®Response: Thank you for your comment. In the top 20 pathway lists (Table 3), after excluding the top 20 pathway terms (Figure 3) and filtering out the common pathways shared by both hypo- and hyperDMGs, we identified pathways that serve as upstream regulators of thymus function in immune regulation, as discussed in (Line 376-443). For hypoDMGs, these pathways include Ras, phospholipase D, cortisol, aldosterone, and growth hormone. For hyperDMGs, the key pathways identified are the Wnt signaling and Fc gamma R-mediated phagocytosis.

Comment #11.

Line 828: In the discussion, it is imperative to mention which genes in the LN group were silenced by epigenetic mechanisms and were not significantly expressed, and how these genes affect the signaling pathway related to the immune system or the growth of the fetus.

®Response: Thank you for your comment. Based on the results of the KEGG pathway analysis (Table S2), we observed that SOS1, a key component of the Ras signaling pathway, and STAT3, a pivotal element in cytokine signaling, are associated with the synthesis and secretion of growth hormones, as well as the proliferation of thymocytes and immune system development. FGF10 and IGF1R are linked to the Ras and Rap1 signaling pathways, which are crucial for cellular growth and organ development, with IGF1R specifically mediating the effects of growth hormones. ADCY1, involved in the cAMP signaling pathway, acts as an inflammatory mediator, while MAPK3 (ERK1), part of the MAPK/ERK pathway, is implicated in cAMP signaling, Ras signaling, and Fc gamma R-mediated phagocytosis, regulating cell division, differentiation, and survival. Although these genes were hypomethylated, which typically suggests the potential for increased transcription, they did not exhibit changes in expression under conditions of maternal nutrient restriction. This suggests that their regulation may involve complex epigenetic or transcriptional mechanisms, possibly as an adaptive response to ensure fetal survival under nutrient stress.

In addition, SLC38A1 is a sodium-coupled amino acid transporter essential for cell growth, particularly in supplying glutamine for immune cell proliferation and fetal development. It is linked to glutamatergic and GABAergic synapse pathways (line 251, Figure 6). Hypermethylation and downregulation of SLC38A1 may impair protein synthesis and cell growth, potentially leading to compromised thymocyte development, a weakened immune system, and broader fetal growth retardation under nutrient restriction. We have mentioned these key genes to the discussion part (Line 416-443). We added the following sentences: “Despite these methylation changes, no significant differences in gene expression levels were detected, suggesting that these genes might remain transcriptionally inactive under MNR. In contrast, SLC38A1 is crucial for glutamine supply in immune cell proliferation and fetal development [39], is linked to glutamatergic and GABAergic pathways (Figure 6). Its hypermethylation and downregulation may impair protein synthesis and cell growth, potentially compromising thymocyte development, immune function, and overall fetal growth under nutrient restriction”(Line 433-440) and deleted the following: “thereby influencing cellular function within the fetal thymus”(Line 440).

I would like to thank Reviewer B for your helpful comments and hope that the revised manuscript is acceptable for publication in the International Journal of Molecular Sciences.

Yours sincerely,
Takafumi Gotoh

Field Science Center for Northern Biosphere, Hokkaido University, N11W10, Kita, Sapporo 060-0811, Hokkaido, JAPAN

Tel & Fax: +81-11-706-3940
